# Design of the Refurbishment of Historic Buildings with a Cost-Optimal Methodology: A Case Study

**José Sánchez Ramos [1]**, **Servando Álvarez Domínguez [2]**, **MCarmen Pavón Moreno [2]**, **MCarmen Guerrero Delgado [2]**, **Laura Romero Rodríguez [2]** and **José Antonio Tenorio Ríos [3,\*]**

1   Department of thermal machines and engines, University of Cadiz, 11001 Cadiz, Spain
2   University of Seville, 41004 Seville, Spain
3   Spanish National Research Council (CSIC), 28033 Madrid, Spain
\*   Correspondence: tenorio@ietcc.csic.es; Tel.: +34-91-302-0440 (ext. 870240)



**Featured Application: The authors are encouraged to provide a concise description of the specific application or a potential application of the work. This section is not mandatory.**

**Abstract:** The transformation of existing buildings into Near Zero Energy Buildings or even positive energy buildings remains a major challenge. In particular, historic buildings are an important cultural heritage that, in most cases, may be rehabilitated and reused for new purposes. However, achieving higher efficiencies in those buildings presents many difficulties, since there is a need to preserve aesthetic values and minimize impact on the buildings' initial construction. In this work, a roadmap that allows rehabilitating a building from the eighteenth century is developed, turning it into a landmark building, to be used as a museum in the near future. The procedure is based on 3D models using REVIT software and detailed energy simulations supported by a cost-optimal methodology. The results reveal how conventional methodologies shown in the literature may improve the energy performance of the buildings during the heating regime, but performance may deteriorate during the cooling season. For that reason, the present study includes the design of a night ventilation system which allows not only solving this problem but also to reducing the cooling demands by more than 43% with little additional costs. In conclusion, historic buildings (which traditionally have a high thermal mass) have increased thermal storage potential by using the structures of the buildings themselves as well as passive cooling techniques.

**Keywords:** near zero energy building; life cycle cost; rehabilitation interventions; night ventilation; passive cooling

## 1. Introduction

### 1.1. Context

The European Directive on energy efficiency in buildings (Directive 2018/844/EU) [1] introduced the need to transform new buildings into Near Zero Energy (NZEB) by 2020. However, existing buildings represent the majority of the building stock, and the transition towards achieving a balance between their energy consumption and production (or even buildings with a larger energy generation than that required for their particular use) presents an interesting challenge. In particular, in the case of historic buildings, this issue becomes more significant given their relevant cultural value, usually limited by environmental, architectural and artistic restrictions, and governed by specific regulations. Therefore, apere is a great need to propose a replicable methodology to improve the conditions of historic buildings so as to achieve sustainability. Kamari et al. [2] demonstrate in their work that



sustainability criteria play an important role in decision-making procedures for refurbishing buildings, since they may integrate all the stakeholders involved in the process.

On the other hand, methods based on the integration of energy efficient solutions with renewable energy technologies will be addressed in this study. For this task, the limitations imposed by architectural preservations will be considered, taking advantage of the constructive peculiarities to improve the buildings.

### 1.2. NZEB

The standard commonly used in Europe regarding this topic is ISO 50001 [3], focused on improving the energy performance of buildings including energy efficiency, use and consumption. This norm establishes the use of an energy management system within existing buildings. This system is founded on the premises of finding potential energy savings with the application of improvement measures and monitoring such improvements from the building's baseline. Hence, the key element is to obtain real indicators of the building and compare them with the reference values. Using the energy savings as benchmark, the selection of the set of measures (catalog defined for the building) should be analyzed.

The developed methodology must be completed by studying all possible improvement measures that make up that catalog. For this, the most used are cost-optimal procedures as proposed in European regulations [4]. In this line, there are many works that have in common the use of the optimal cost procedure [5–8]. The main objective of these studies was the evaluation of the interest of every measure and their combinations, allowing finding the optimal alternative by taking into account different constraints: economic optimum, maximum extra cost or maximum payback period.

Another point of view of the procedure to obtain NZEB appears in the standard [9]. This standard defines the first three stages to achieve net zero energy buildings. The first stage requires knowing the real needs of the building and acting on them (step 1: energy needs). Once characterized, the need to analyze the systems of that building arises (step 2: energy consumption). In addition, finally, overcoming these two points becomes imperative to analyze the possibility of producing the energy consumed by that building (step 3: renewable energies) or even producing more and exporting the surpluses. In addition to the aforementioned challenges, there is an intrinsic complexity of acting on existing buildings (rehabilitation).

If the standard in [9] is analyzed, it may be understood that stages 1 and 2 are linked to the difficulty of accurately characterizing the real energetic situation of the building. Related to this, there are many research studies that base their results on the estimation of the needs of the buildings, for example studies on the implementation of renewable energies. The majority of these works show the need to carry out a realistic modeling of the building, both geometrically and energetically. Romero et al. [10–12] developed different approaches to model buildings with the available geographical data: the complete study of photovoltaic potentials with 3D district modeling [10], thermal loads simulation [11] and shading analyses for studies related to solar resources, also defining operation strategies [12]. Presently, studies on the estimation of PV potentials are relevant and interesting, as shown by Costenzo et al. [13].

The roadmap design to obtain an optimal rehabilitation project was previously defined by life cycle studies, appearing in research and norms developed from several studies [14–16]. These studies can be classified according to the procedure implemented for the evaluation of the energy performance: detailed procedures with commercial or own simulation tools, and simplified procedures.

### 1.3. Procedures and Methodologies

From the review of the studies linked to detailed procedures, we may highlight that detailed results can be achieved with a short simulation time step. Many studies such as [17–29] present detailed sensitivity analyses in the variations of their parameters, and even hourly analyses of the results. Also, the high computational cost derived from the use of tools such as EnergyPlus [30] or TRNSYS [31] and the simulation of thousands of cases should be mentioned. The most widespread method is the use of cost-optimal procedures based on massive simulations from different existing alternatives [22,32–34].

There are also studies in which the search for the optimal solution was found through the design of multivariable optimization procedures for decision-making and simulation of cases in a directed manner, for example in [6,33,35–37].

The studies based on simulations with detailed tools allow studying energy saving measures in detail, but they require high computational expense and hours of work to prepare the cases. Additionally, the decision-making process involves the need to adapt or develop complex multivariable optimization methods to direct the search for the optimal project. However, this methodology has been commonly used in previous literature.

Several methodologies for the performance evaluation and optimization of existing buildings were proposed, as can be seen in [38–42]. In particular, the use of dynamic simulation tools was identified as adequate to predict the thermal energy performance of buildings [43,44]. Therefore, dynamic simulations together with experimental measurements and sensitivity analyses could improve substantially the optimization of the building operation process towards energy efficiency.

In recent years, several design options and measures were detected in heritage buildings to improve energy efficiency and reduce energy demand, as well as the buildings operating costs [45]. The main step generally carried out consists on energy audits, evaluating the building's energy performance [46]. In particular, the implementation of internal insulation for the thermal envelope, cold coatings and improvement of the windows represent the best solutions to optimize the buildings' energy efficiency. Additionally, other measures are proposed, such as control systems, lighting or ventilation, along with heat recovery systems. All the proposed alternatives are listed as the main technologies to reduce energy demand in heritage buildings with temperate-Mediterranean climatic conditions [47].

The level of scientific and technical development related to building rehabilitation is high, raising interest around the subject as demonstrated by Jenkins et al. [48]. In turn, Filippi et al. [49] established as historical buildings those built prior to 1945, and proposed a series of indicators to stimulate their renewal. The results of this work show that historic buildings are equally affordable when compared to conventional buildings in need of energy rehabilitation, although its heritage aspects must be taken into account. Bertolin et al. [50] present a decision-making methodology to prepare action projects on historic buildings, and conclude with the need to address the thermal insulation of the envelope. The thermal envelope has been a protective mechanism against the climatic and anthropogenic influence in buildings with a specific use, acting as a key element for any rehabilitation intervention.

On the other hand, there is a need to preserve aesthetic values, especially in unique buildings such as museums or churches. In Pachta et al. [51], an in-depth study of materials was carried out to perform an intervention on a religious building, minimizing the impact on the building's initial construction. The work showed the challenging process of the study, and the positive implications of improvements in these types of construction for the long-term preservation of buildings. As the authors Santos et al. [52] insist, the aesthetics of the buildings (a parameter not allowed to be modified by United Nations Educational, Scientific and Cultural Organization (UNESCO) is their cultural essence, as well as the primary element in maintaining tourist interest. Yüceer et al. [53] showed within their research the importance of the buildings' exterior, highlighting the envelope from historic assets and additionally presenting a methodology to update and improve the buildings' envelope through current rehabilitation techniques and innovative materials.

### 1.4. Aims

Given the existing issues and the facts presented above, the purpose of this study is to define a replicable methodology for buildings' energy rehabilitation, particularly those that are considered to be part of the historical heritage of cities. This will be done by taking advantage of the specifications of different construction techniques from the past with a vision towards the future, using an energy efficiency perspective for the environmental preservation of ancient urban centers. The methodology

proposed is based on finding the optimal refurbishment alternative by using a cost-optimal method, analyzing independently the heating and cooling demands.

This work will thus develop a roadmap that could allow historic buildings to be renovated, as well as illustrate the improvement of the conditions of an emblematic building. The case study raises a conflicted issue due to its location (Cádiz, Spain, Mediterranean zone) and the fact that the cooling regime is more pronounced than the heating regime. Therefore, there is a need to innovate with respect to existing trends found in the literature so as to avoid the overheating problem. The building was originally used as a convent, later as a school and potentially will be used as a museum with some offices in the near future, with almost zero energy consumption.

## 2. Methodology

### 2.1. Overview

The methodology presented in this work has the objective of proposing a replicable method for the energy rehabilitation of buildings that belong to the historic patrimony of the cities where they are located. Figure 1 shows the different stages that should be carried out during the course of energy rehabilitation interventions.

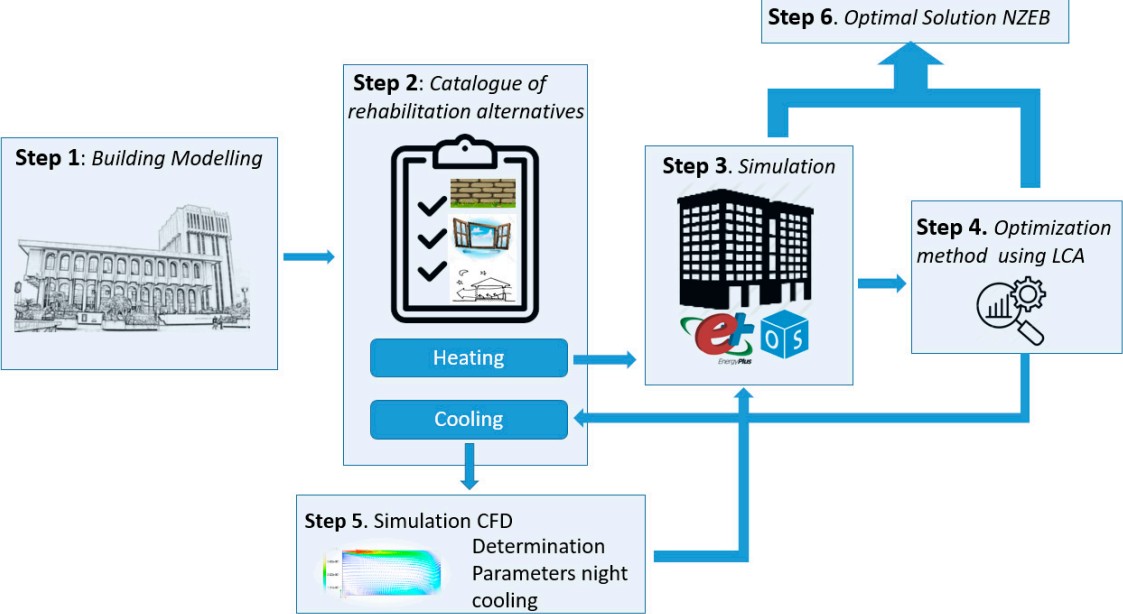

**Figure 1.** Overview of the methodology. CFD: Computational Fluid Dynamics; NZEB: Near Zero Energy; LCA: Life cycle assessment.

Each of the stages shown in Figure 1 encompasses a series of key work actions. The first phase considers an accurate study of the building based on BIM (Building Information Modeling) standards, using the REVIT software [54] in order to develop a model. In the second stage, the catalog of conventional measures applicable to the building in question is defined. The rehabilitation alternatives proposed seek to reduce the heating and cooling needs. This second step is connected to the next stage by the need to evaluate the energy impact of these measures through dynamic simulation using EnerPlus [30]. The results from the third stage feed a decision-making procedure (stage 4) based on Life Cycle Cost analysis (LCC). The result from the fourth stage is the optimal package of solutions. The problems related to the cooling regime were analyzed. As stated in the introduction, conventional measures are aimed at reducing heating needs, as in the PassivHaus standards [55].

As an innovative addition to the methodology, the design of an optimal night ventilation strategy was implemented as the means for the proper application of passive cooling strategies in the case of

the present work, accounting as the fifth step with the goal of improving the decision-making process. Finally, in the sixth stage, the cost-technical details (among other variables) that would certify the viability of the energy rehabilitation of the building towards a NZEB are checked.

The final aim of steps 2, 3 and 4 in Figure 1 is to propose a catalog of measures for the buildings' energy rehabilitation, considering the heating season to begin with, through the application of measures that could improve the thermal envelope, reduce the thermal bridges and the air infiltrations. This should be done without neglecting the resulting demands for the cooling season, since they may be affected by the increase of insulation of the building. In addition, there are other alternatives to improve it a posteriori, such as by using solar control and night ventilation. The question that is raised is the choice of the optimal combination which would mean a notable decrease of the energy demand, as well as the lowest possible life cycle cost. For this, the optimal cost methodology established by the European Directive was used [5,7].

## 2.2. Building Modeling

The case study is a building from the eighteenth century located in the old town of Cádiz, known as the Rosario Institute, surrounded by other buildings with similar historical and cultural value. During much part of its history, the building belonged to the Catholic Church as a convent formerly known as "San Agustín". It was reformed for the last time in 1995, changing its classification to a tertiary building, as it is currently known. The building has four floors with a rectangular interior patio, constituting a total constructed area of 2500 m$^2$ as is seen in Figure 2.

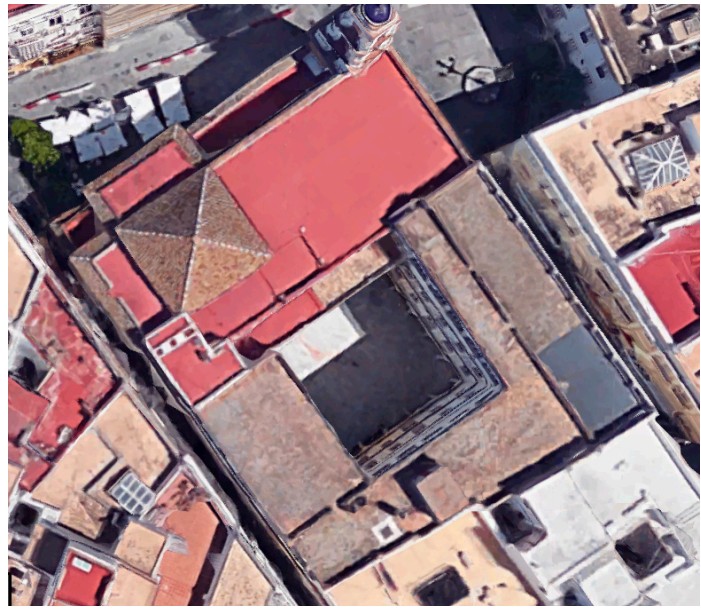

**Figure 2.** Aerial view of the building in the case study.

First of all, in order to carry out the analysis for the building's improvement, it is necessary to establish the starting point. For this, it is important to model the building as close to the reality as possible, both regarding the constructive characteristics as well as the buildings' operation. In this way, the main parameters of the building, used for the definition of the catalog of alternatives and the subsequent simulations, are obtained.

To carry out such a model of the case study, a visit to the building was made with the objective of making a visual inspection to describe the current status, and thus have a better idea of the situation and condition of all the elements. The condition of the enclosures was verified, as well as the distribution of spaces in each floor, windows, solar access, location of the walls (or parting walls that serve as a boundary between two properties), and the exterior walls' thickness (those facing the street as well

as those facing the courtyard inside). That information was complemented with that provided by both the buildings' drawings and the building model obtained using REVIT [54] (see Figures 3 and 4), which allowed defining the geometry of the building including the space partitions, the empty spaces, the different floors and the elements' constructive materials.

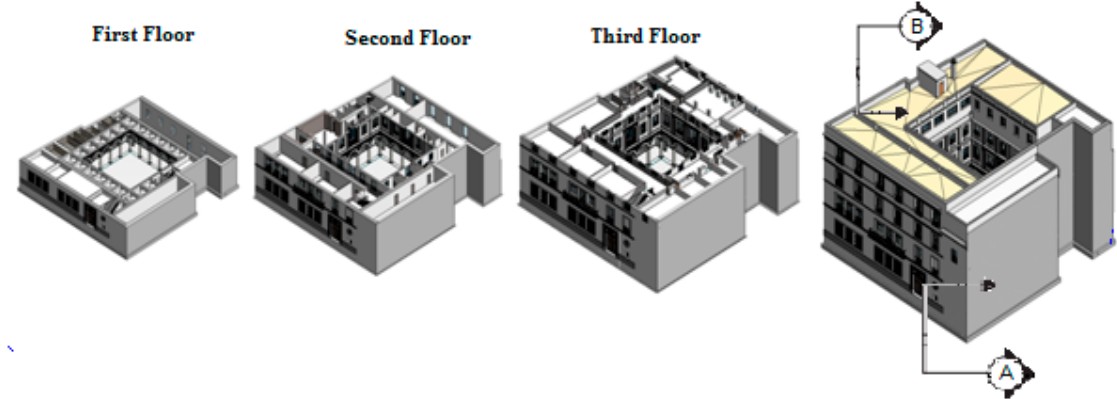

**Figure 3.** 3D model of the building, isometric view.

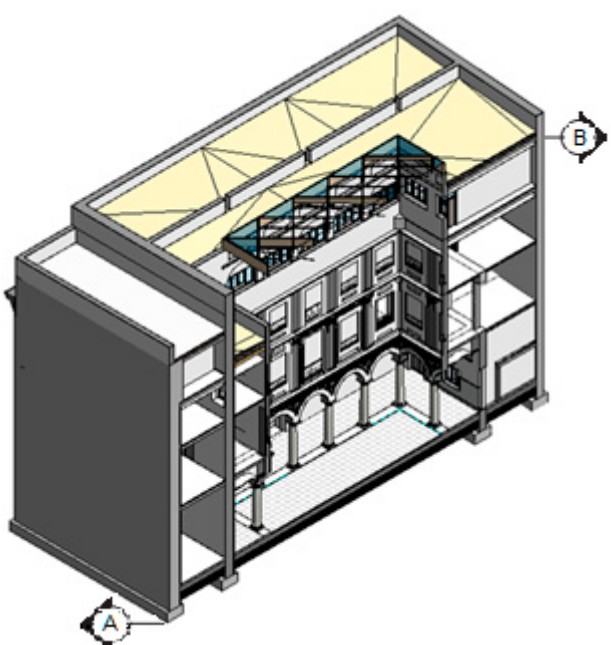

**Figure 4.** 3D cross section of the building.

The building consists of four floors with an inner courtyard, as shown in Figures 3 and 4. All floors have a similar distribution. Regarding the constructive characteristics, as can be seen in the model, the facades (including both the facades facing the exterior and the ones facing the internal courtyard) are constituted of natural stone covered with mortar and finished with paint (see Figure 5). The model also includes the brick interior partitions, wood carpentry, simple glass doors and windows (shown in Figure 5-right) and finally a flat waterproof roof without insulation.

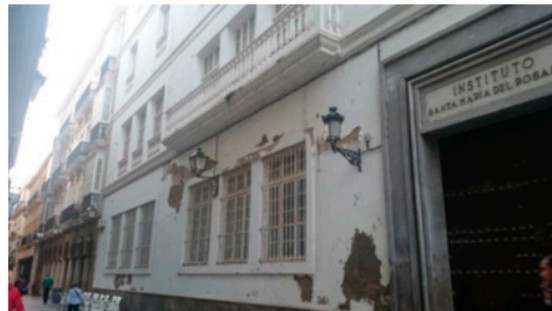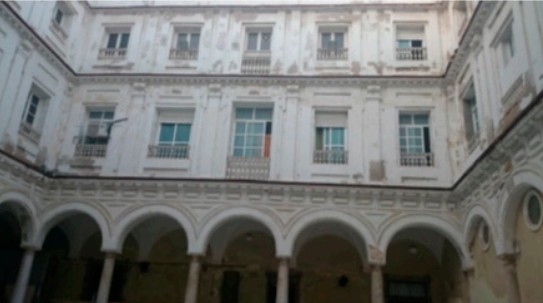

**Figure 5.** Facade of the building facing the street (**left**) and the inner courtyard (**right**).

Initially, it is important to determine the constructive characteristics such as materials and thickness of the different layers of the wall layout (Appendix A). Table 1 shows the thermal transmittance of each of the buildings' constructive elements. This was done in order to make sure that the model of the building represents the actual construction in the most reliable possible way.

**Table 1.** Thermal transmittance of the building's constructive elements.

| U-Value [W/m²·K] | Exterior Walls | Interior Walls | Roof | Floor | Windows |
|---|---|---|---|---|---|
| | 1.21 | 1.64 | 3.31 | 3.32 | 5.70 |

Regarding the building's facilities, as it is an historic building it does not have an air conditioning system. For this reason, installations from classic references will be considered: the generation of hot water for heating with a condensing boiler and cold water with a chiller. In the same way, all the remaining infrastructures such as Domestic Hot Water (DHW), lighting and ventilation are not operating under optimal conditions. Therefore, reference systems were also chosen for the simulations.

An operation time of 8 h during the day was selected, time during which the air conditioning equipment will work, specifically from 7:00 a.m. to 3:00 p.m. During this time interval, the heating and cooling set-point temperatures will be set to 20 and 25 °C respectively. Outside of this schedule, the building will be in free floating mode.

Finally, the mechanical ventilation of the building will be activated during the occupation period, establishing 1.6 ren/h, as stipulated in the Regulation for the Thermal Installations in Buildings in Spain (RITE) [56] when taking into account the current use of the building and its occupation. Outside of this period there will be infiltrations from the building itself.

The model of the building was generated using the DesignBuilder tool [57], performing the energy calculations through the EnergyPlus dynamic simulation software [58]. The input parameters for the simulation are presented in Table 2.

**Table 2.** Main input parameters for the dynamic simulation of the case study building.

| Input Parameters for the Simulation | |
|---|---|
| Location | Cádiz, Spain |
| Use | Museum |
| Total Area [m²] | 2534 |
| Total Volume [m³] | 11150 |
| Number of floors | 4 |
| Occupancy level [people/m²] | 0.5 |
| Internal gains [kwh/m²/year] | 2500 |
| Occupancy schedule | 7:00 am–3:00 pm |
| Infiltration level [1/h] | 0.75 |
| Current HVAC technology | Reference installation |

The simulation of the building in its original state allowed the calculation of the heating and cooling demands shown in Table 3, which help to define the catalog of proposed improvement measures and the subsequent optimization method.

**Table 3.** Heating and cooling demand of the building.

|  | Heating Demand [kWh/m$^2$] | Cooling Demand [kWh/m$^2$] |
| --- | --- | --- |
| Original situation of the building | 24.66 | 21.55 |

### 2.3. Conventional Catalog—Heating Needs

The problem of the current building, which has already been analyzed in detail, requires that the building be rehabilitated through actions that improve the quality of the envelope, the thermal bridges or the building's infiltration. However, there are countless possibilities, each of which bear costs associated with a decrease in building energy needs for heating and cooling. The question that arises is the choice of the optimal combination, which would imply a notable decrease in the demand, as well as the lowest possible life cycle cost. All the alternatives of improvement proposed for each of the components are described below:

- **Enclosures.** Improvements in the transmittances of walls, roofs and floors. The application of these improvements is based on the placement of thermal insulation inside the building (so as not to modify the external appearance of the construction) and the use of suitable materials. In addition, a distinction is made between the exterior walls that face the street, and those that face the interior courtyard. In this way, it is possible to know if it is necessary to improve both types of walls, or if it is necessary to give priority to the walls facing the street.
- **Windows.** The replacement of the existing single glass windows to double glazing with an interior air chamber is proposed, with a transmittance reduction that can reach 50% in some cases, depending on the thickness of the chamber.
- **Thermal bridges.** The improvement of the building's thermal bridges will also be considered, since these can contribute greatly to the demand of the building. The considered scenarios propose a reduction of up to 40%.
- **Infiltrations.** In the case of ventilation, during the periods of occupancy the values defined in the operational conditions of the building will be maintained. However, outside the periods of occupation different infiltration levels will be considered, starting from 0.75 Air Changes per Hour (ACH) until reducing it by one tenth.

Therefore, after considering all the previously defined alternatives, the following table shows the catalog of measures, depending on the transmittances achieved and the changes associated with the proposed rehabilitation. The first row of Table 4 corresponds to the current situation of the building. The lowest possible values are those recommended by the Technical Building Code in Spain depending on the climatic zone, in this case A3, named as High Efficiency values. Taking into account all the options, there is a total of forty-five thousand (45,000) possible combinations.

Due to the high number of alternatives proposed, different levels of improvement were considered to simplify the problem. In total, 54 combinations (see Table 4, yellow zone) were analyzed in depth: six different levels of enclosure improvements, three levels for the thermal bridges and three infiltration levels. The chosen values constitute a representative sample of all the alternatives originally proposed.

It is important to say that infiltration rates in tertiary buildings are lower than in residential buildings [59], and the ratio of infiltration rates and space volume is small. For these reasons, Table 4 considers two different levels of infiltration rate. It should be noted that the studied conventional alternatives are the ones recommended by the Energy Performance Building Directive EPBD [60], particularly the passive techniques that affect the quality of the thermal envelope of the building, as shown in the report in [61].

**Table 4.** Catalog of conventional measures for the buildings' constructive elements. ACH: Air Changes per Hour.

| U-Value [W/m²·k] | | | | | Length Reduction | Air-Flow [ACH] |
|---|---|---|---|---|---|---|
| **Exterior Wall** | **Interior Wall** | **Roof** | **Floor** | **Windows** | **Thermal Bridges** | **Infiltrations** |
| 1.21 | 1.64 | 3.31 | 3.32 | 5.7 | Actual | 0.75 |
| 1 | 1.25 | 0.8 | 0.8 | 2.6 | 10% | 0.5 |
| 0.8 | 1 | 0.47 | 0.53 | 2.1 | 20% | 0.25 |
| 0.5 | 0.8 | 0.3 | 0.4 | | 30% | 0.075 |
| 0.3 | 0.5 | 0.23 | 0.37 | | 40% | |
| | 0.3 | | | | | |

The aforementioned alternatives generate a battery of cases with all their possible combinations. These cases were simulated with EnergyPlus so as to be able to estimate the energy consumption, the life cycle cost and the different energy savings with respect to the buildings' original situation.

*2.4. Optimization Procedure*

For the search of the optimal improvement alternatives, the optimal cost methodology established by the European Directive was used [5,7]. The simulations carried out allowed the comparison between the different improvement alternatives. For this purpose, the LCC was estimated for a period of 30 years using Equation (1), which would imply taking into account the costs associated with the alternatives from the heating catalog (changes in the envelope, thermal bridges and infiltrations), as well as the operating costs for the air conditioning equipment in each case.

$$\text{LCC} = \text{Cost of improvement alternatives} + \text{Operation cost} \cdot \sum_{k=1}^{30} \frac{1}{(1 + 0.02)^k} \qquad (1)$$

Table 5 shows the extra costs associated with the proposed improvements, which were implemented in the calculation of the LCC. Rt is considered as the increase of thermal resistance, the quotient between the thickness of the insulation incorporated in each case and its conductivity, chosen as 0.036 W/mk.

**Table 5.** Extra costs associated with the improvements proposed.

| Heating Alternatives from the Catalog | Costs | Units |
|---|---|---|
| Insulation: walls | 2.73·Rt + 2.01 | €/m² |
| Insulation: roofs | 1.70·Rt + 1.95 | €/m² |
| Insulation: walls | 4.14·Rt + 2.38 | €/m² |
| Windows U = 5.7 | 182 | €/m² |
| Windows U = 2.6 | 228.26 | €/m² |
| Windows U = 2.1 | 246.15 | €/m² |
| Treatment of thermal bridges (30%) | 6.0 | €/m |
| Treatment of thermal bridges (40%) | 7.0 | €/m |
| Infiltrations | 60 | €/m² |

As can be seen in the previous table, all the alternatives proposed in the catalog correspond with measures that increase the insulation-tightness of the building in general. This causes greater cooling demands. For this reason, there is a need to analyze those optimal candidates for the heating regime improvement process that do not increase the cooling demand by more than 30%. This is done with the aim of improving the alternatives to the cooling catalog and achieving the desired situation of buildings with balanced energy needs.

### 2.5. Extension of the Catalog—Cooling Needs

Once the optimal solutions to improve the building during the heating season have been identified, the improvement analyzed implies that the energy demand for the cooling season would be higher, due to the increased insulation of the building. This requires solutions to reduce the demand during summer. The suggested alternatives will be night ventilation and solar control.

The most frequently studied alternatives associated with solar control correspond to both fixed and mobile solar protections, solar controlled windows and innovative ventilated facades. These measures are especially interesting for buildings with high solar incidence: most of them are low cost technologies and achieve significant savings. However, the studied building has limited solar access. Therefore, the use of solar control measures due to the conditions of the building and its surroundings has not been accounted for.

On the other hand, ventilation strategies focus on exploiting the potentials of free cooling as an important alternative to reduce cooling demands. They are widely studied and one of the most promising passive cooling techniques [62]. If the outdoor air temperature at night is low enough, ventilation can be used to cool the exposed thermal mass of a building. In this way, free cooling through optimized night ventilation could be an efficient technique to reduce the cooling demand of the building.

Most studies conclude that the use of night ventilation in buildings can lower the maximum indoor temperature of the next day up to three degrees. In parallel, when applied to buildings with air conditioning, a considerable reduction in the cooling demand can be expected [63]. However, most studies refer to commercial buildings, very few of them refer to the application of night ventilation in residential buildings [64–66] and almost none to historic buildings. In this way, the use of night ventilation techniques is proposed in this work as a possible alternative to reduce the demand for cooling in heritage buildings, since apart from achieving such a reduction they do not affect the appearance of their architecture.

For the study, we proceeded to calculate the demand of the building (in its original state) and the cases chosen as optimal for different combinations of night ventilation (using a mechanical system): 0, 4, 8, 12 and 16 ACH, which will be activated for nine hours starting at 9 pm. Night ventilation measures will take place only as long as the outside temperature is lower than the indoor temperature of the building and the schedule is the appropriate (21:00–6:00 h).

To carry out these simulations, it was necessary to use numerical simulation software, specifically, ANSYS Computational Fluid Dynamics (CFD) Fluent [67], with the objective of obtaining the efficiency of the heat transfer process, defined in Equation (2) as an intermediate step to calculate the convection coefficients. These coefficients together with the introduced air flows and the inlet temperatures allow implementing the alternatives in the simulation software EnergyPlus, as well as carrying out the optimization process, which was previously explained, in a similar way. In addition, with this CFD analysis, it is possible to undertake an in-depth study of the possible distribution of the air extractors that should be installed, the behavior of the air that is introduced, as well as its movement around the studied area.

The efficiency process was defined by treating the night ventilation process as a heat exchanger, where the transfer processes take place between the inlet air (with a lower temperature) and the walls, especially those named as "massive walls" due to their higher mass among the area under study.

$$\varepsilon = \frac{m \cdot cp \cdot (T_{inlet} - T_{outlet})}{m \cdot cp \cdot (T_{inlet} - T_{wall})} \tag{2}$$

The numerator in Equation (2) corresponds to the real heat transfer, which may be formulated in the following way since it is a convection phenomenon:

$$Q_{cv} = m \cdot cp \cdot (T_{inlet} - T_{outlet}) = h \cdot A \cdot (T_{inlet} - T_{outlet}) \tag{3}$$

This heat in Equation (3) is capable of obtaining a reduction of the cooling demand (see Equation (4)), where the efficiency obtained in Equation (2) intervenes. As a result, all the parameters for the calculation of the convection coefficient on any wall are known.

$$\nabla Cooling\ Demand\ =\ \rho \cdot cp \cdot V \cdot ACH \cdot DD_{night} \cdot \varepsilon \tag{4}$$

In this way, it is possible to obtain the convection coefficients that will be introduced in the building simulation software, with the objective of calculating the demands of each combination and apply again the optimization criteria. The introduction of these night ventilation alternatives does not influence the heating demands in any way, but the cooling demands could be greatly diminished thanks to the ability of providing free cooling with no additional costs but the one of the ventilation system, which introduces cooler air from the exterior of the building.

## 3. Results

### 3.1. Results of the Conventional Catalog

First of all, according to the proposed methodology, the results of the application of the alternatives from the catalog associated with the improvement of the heating demand are analyzed, which are the alternatives for the rehabilitation of the building envelope. This conventional catalog reduces considerably the heating demand, as shown in Figure 6.

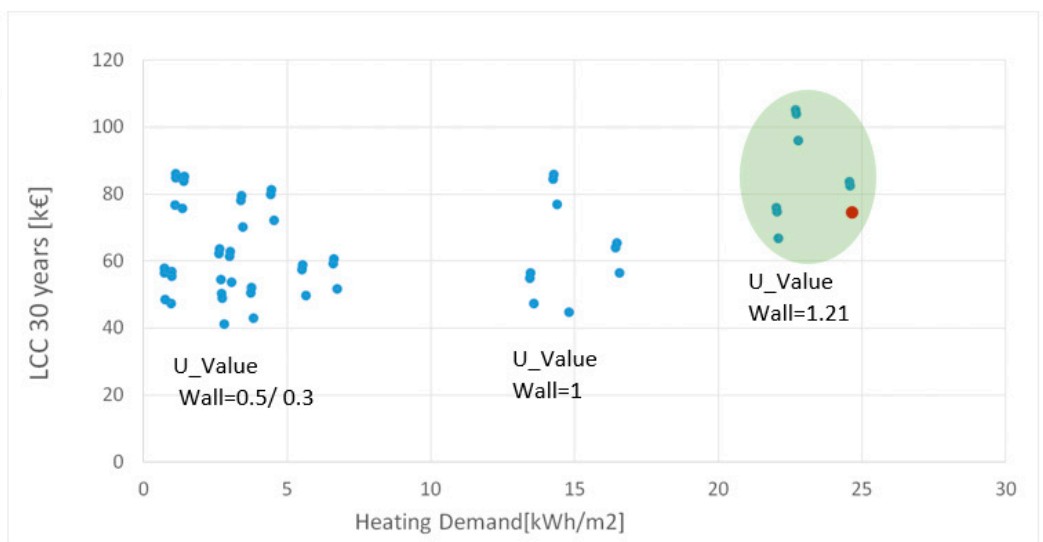

**Figure 6.** Life cycle cost results for heating demand—conventional catalog.

In Figure 6, it can be seen how the demand for heating is diminished as the transmittance of the different elements is reduced, with respect to the starting position (red dot). Then, we proceed to analyze the influence of the different alternatives according to their greater or lower effect to the reduced amount of heating demand. The points closest to the initial position, shown as the green shaded area, correspond only to the thermal bridge improvement alternatives, which have a limited effect on the improvement of the demand, mainly due to the low quality of the envelope in the original case. The rest of the alternatives (improvement of the envelope with increased insulation), suppose important reductions in the heating demand, especially with the reduction of the transmittance from the envelope, particularly in walls.

Taking into account all the combinations, the steps of the methodology have been observed in the different values of transmittance of the walls, which are shown in Figure 6. However, these measures

imply a notable increase of the cooling demand (see Figure 7). This is due to the fact that higher insulation prevents the dissipation of the heat accumulated inside the building during summer.

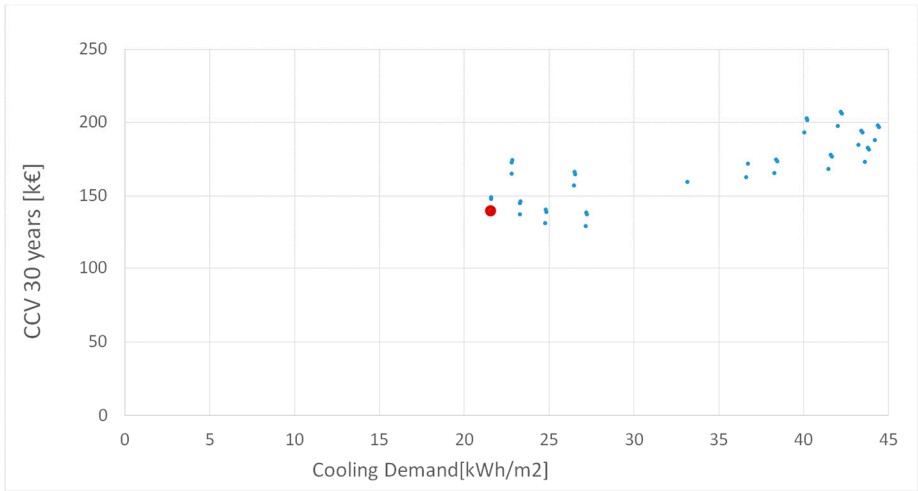

**Figure 7.** Life cycle cost results for cooling demand—conventional catalog.

The next step will be to detect those combinations that can be considered to be optimal candidates. First, the points closest to the lower left corner (yellow shaded area shown in Figure 8) were chosen as optimal, since they achieved the best demand for heating and at the same time reach the minimum life cycle cost. However, those cases correspond in Figure 7 with the greatest values of cooling demand, since they imply a high level of insulation. For this reason, the alternatives shown in red should be chosen instead, since they increase the cooling demand less than 30% (33.14 kWh/m$^2$) and therefore allow a margin of improvement later in cooling. Table 6 justifies why the optimal case guarantees greater benefits in the cooling regime.

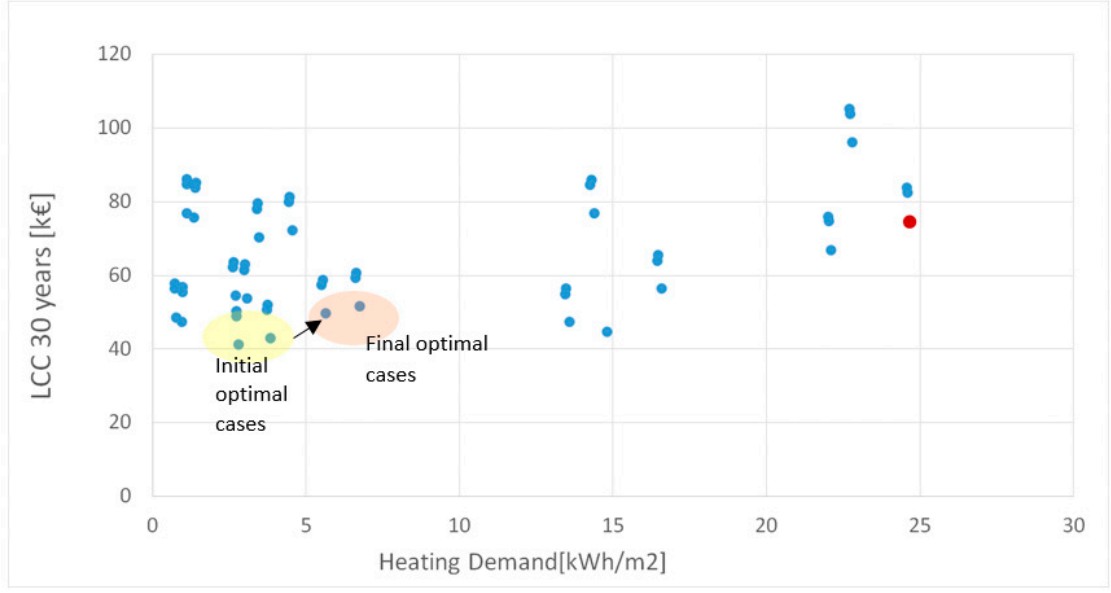

**Figure 8.** Analysis of life cycle cost results for heating demand—conventional catalog.

**Table 6.** Description of the optimal results in the heating mode.

| Case | U-Value Inner wall [W/m²·K] | U-Value Outer wall [W/m²·K] | U-Value Roof [W/m²·K] | U-Value Soil [W/m²·K] | U-Value Window [W/m²·K] | Thermal Bridges | Infiltration [1/h] |
|---|---|---|---|---|---|---|---|
| Initial scenario | 1.21 | 1.64 | 3.31 | 3.32 | 5.7 | 0 | 0.75 |
| Optimal scenario | 0.5 | 1.00 | 0.47 | 0.53 | 2.6 | 0 | 0.75 |

Finally, Table 7 compares the demands of the buildings' original case with the selected improved case. It can be observed that the cooling demand increases by 70%, while the heating demand is reduced by 73%.

**Table 7.** Comparison between the initial situation of the building and the optimal heating alternative.

| | Heating Demand [kWh/m² Conditioned] | Cooling Demand [kWh/m² Conditioned] |
|---|---|---|
| Initial scenario | 24.66 | 21.55 |
| Optimal scenario | 5.73 | 33.14 |

### 3.2. Results of the Advanced Catalog

Once the optimal heating combination is verified, it is clear how the improvement in heating means an increase in the cooling demand due to the building's high insulation, which requires night ventilation or solar control to diminish the cooling demand.

It has previously been stated that the building under study has a low solar access; therefore, the use of solar control measures is not considered and the optimization process will be focused only in reducing the cooling demand with the application of night ventilation.

First, starting with the optimal case from the heating regime with the characteristics presented in Table 8, different combinations of night ventilation are applied: 0, 4, 8, 12 and 16 ACH using the numerical CFD simulation software [67] for the preliminary calculation of the film coefficients, which later allows the simulation of those alternatives in EnergyPlus [58] to calculate the different demands and obtain the optimal case.

**Table 8.** Calculated night cooling efficiencies (Equation (2)) using Computational Fluid Dynamics (CFD).

| Case of Night Ventilation [ACH During Night Hours] | Efficiency (%) |
|---|---|
| 16 | 18.76 |
| 12 | 24.15 |
| 8 | 33.01 |
| 4 | 33.67 |

The explanation of the process that followed is analyzed with the example of one of the modules of the building, an area on the fourth floor, given the similar distribution of all the floors. Figure 9 shows the selected area highlighted in blue. The procedure would be similar for different modules or floors, as well as the building as a whole.

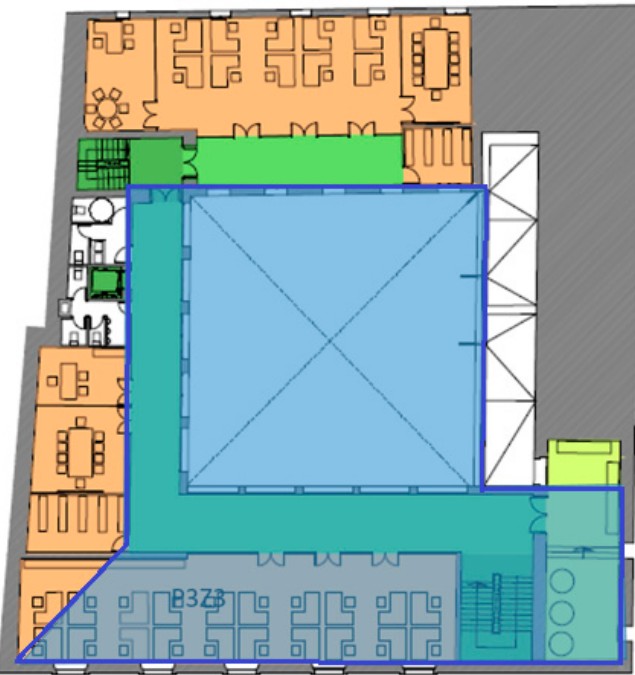

**Figure 9.** Plan view of the top floor of the building.

The enclosure where the air will be circulating was modeled as a solid, and six extraction grids have been identified at the ends of the courtyard, with the aim of choosing their optimal location according to the behavior from the ventilation strategy simulated. This procedure was followed in the case of the windows as well. The wall connecting the interior space is considered a bearing load. The upper floor is considered with the windows of the building itself, the opening of the balconies located next to the interior patio, and in the upper part of the patio end, the extraction grilles (see Figure 10). The wall that connects the patio with the interior space is considered a load wall, in order to use its thermal inertia to improve the comfort conditions indoors.

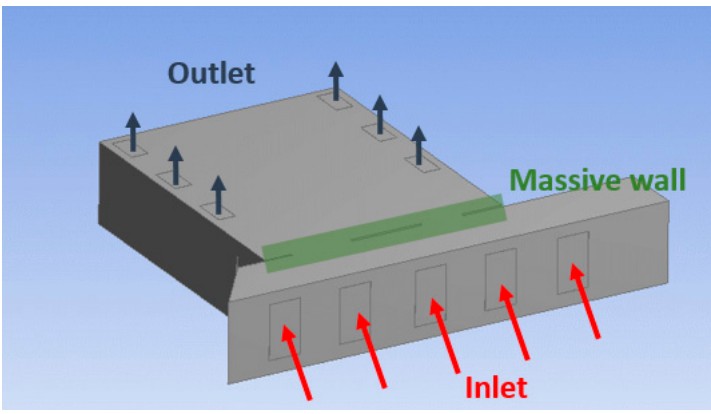

**Figure 10.** Ventilation strategy model for one floor of the building.

The hexahedral mesh was made using ANSYS software v19.2 (Pittsburgh, PA, USA) [68] and refined in the areas near the walls where greater contact with the incoming air is needed, ensuring the accuracy of the results in the calculation of the heat transfer coefficient. The defined mesh guarantees that the yplus [69] of the walls selected for the analysis reaches values close to one. Due to this, the optimal value of this parameter allows analyzing thermal effects near the walls.

Finally, before proceeding with the simulation, it is necessary to determine the boundary conditions in Fluent, considering different locations for the windows to define each ventilation strategy. The input

parameters for the simulation model have been defined as the air flow and an external temperature of 15 °C, using the ventilation grids from the extractor as the outlet. In this way, the film coefficients could potentially be obtained, in order to calculate the energy demands for each combination to apply the optimization criteria again. It is noted that by using these measures of night ventilation the demands for heating have not been affected and in any case. Conversely, the demand for cooling could be significantly reduced, which implies the improvement of the comfort conditions within the building due to the free cooling strategy, consisting of a ventilation system that would allow air with lower temperatures to enter the building. This strategy could potentially be substantially improved by changing the placement of the ventilation grids to the bottom of the courtyard, given that air closer to the ground tends to have lower temperatures.

Once the efficiencies (see Table 8) have been obtained through simulations and taking into account the calculation process defined in Equations (2) and (3), the film coefficients of the walls are obtained. The values for the wall named as "massive" are represented in Figure 11, and depend on the introduced air-flow, considering two air extractors in this case. It can be seen that the film coefficients increase as the air flow rate introduced increases, and reaches an average value of 12 ACH around 10 W/m² K.

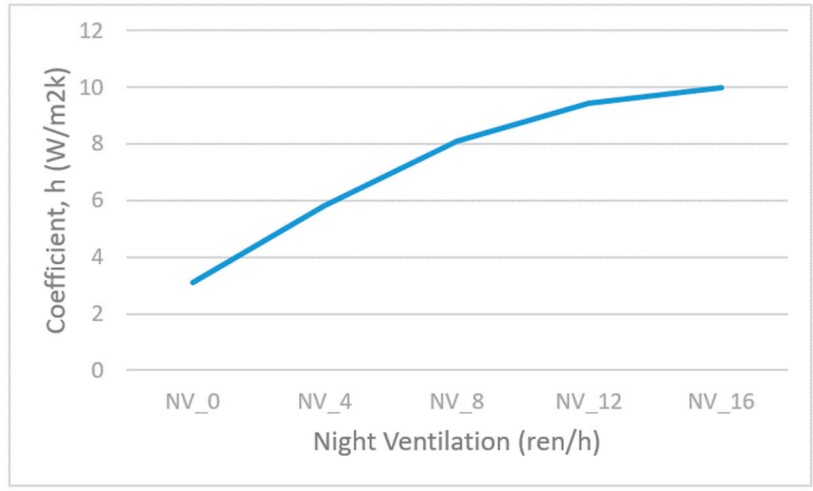

**Figure 11.** Results of heat transfer coefficients (activation of night ventilation).

As a complement to the basic study of the film coefficients, one can observe the results obtained from the simulation program (see Figure 12). The highest speeds are reached both in the wall in question and in the points of extraction. If we zoom in on the analyzed wall, it is possible to observe the movement of the air around that wall.

Once the night ventilation was characterized in order to incorporate it into the simulation program, it is possible to obtain the demands for each of the presented situations. It is verified that by using these measures of night ventilation, the demand for heating is not affected in any case, and the cooling demand on the contrary is significantly reduced, thanks to the cooling effect from the structure of the building. This could be done considering only the consumption of the ventilation system, which involves the use of air flows at temperatures lower than those from the interior of the building.

Following the proposed methodology, the coefficients of convective transfer that appear in Figure 11 are used in EnergyPlus on the case chosen as the optimum from the conventional catalog for heating improvement (see Section 3.1). The results of the night ventilation measures appear in Table 9 for the different air flows that were studied.

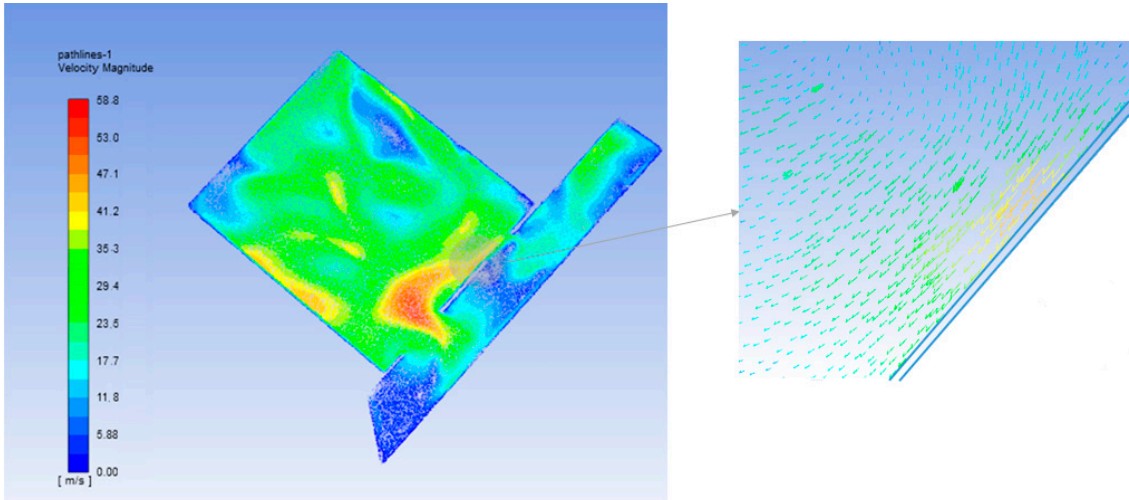

**Figure 12.** Example of CFD results: analysis of flow patterns.

**Table 9.** Final results of improvements.

|  | Heating Demand [kWh/m² Conditioned] | Cooling Demand [kWh/m² Conditioned] |
|---|---|---|
| Initial scenario (see Table 8) | 24.65 | 21.55 |
| Optimal scenario (see Table 8) | 5.73 | 33.14 |
| Optimal scenario (see Table 8) + case of 4 ACH night cooling | 5.73 | 24.78 |
| Optimal scenario (see Table 8) + case of 8 ACH night cooling | 5.73 | 21.42 |
| Optimal scenario (see Table 8) + case of 12 ACH night cooling | 5.73 | 19.76 |
| Optimal scenario (see Table 8) + case of 16 ACH night cooling | 5.73 | 18.73 |

The results shown in Table 9 allow the calculation of new life cycle scenarios for the building. Figure 13 shows the original situation of the building (red point), the optimal situation obtained from the base catalog (NV_0) and the different night ventilation scenarios (NV_ACH during night hours).

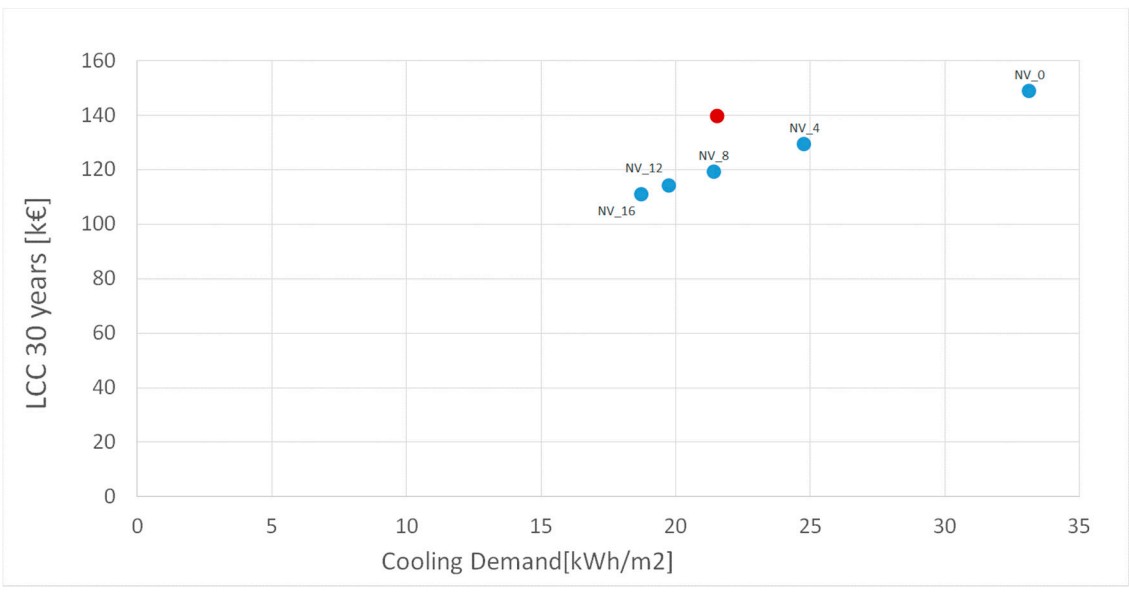

**Figure 13.** Life cycle analysis using the advanced catalog.

Figure 13 shows interesting results regarding the night cooling potential. The difference in the *Y*-axis between the night ventilation strategies and the point NV_0, is the maximum allowable extra cost. That difference between the life cycle costs would be the initial investment cost. However, the estimated initial investment is 12 k€ for the entire building, taking into account the grids, extractors, actuators in windows and control devices.

Finally, the results that have been presented validate the methodology exposed in the previous section, and show the possibility of its implementation in all types of buildings. Thanks to these results, an optimal rehabilitation project was designed that involves the recovery of a historic building.

## 4. Conclusions

Rehabilitating historic buildings is an important step for their long-term preservation, and these buildings present certain construction features (such as high thermal mass) which result in high energy saving potentials. However, the process of achieving standards such as NZEB in historic buildings is very challenging, since the aesthetics of the buildings should not be modified, in order to retain their cultural essence. One way of developing this work, with higher potential in the future, could be the use of more innovative approaches to rapidly generate renovation scenarios, as shown by Kamari et al. [70].

In this work, a methodology for a building's refurbishment is applied to a building from the eighteenth century, which needed an integral rehabilitation. The building is located in a Mediterranean climate, where the cooling needs are not negligible compared to the heating needs. That is the reason, as the present work shows, conventional retrofitting alternatives are not enough to reach an optimal solution. Because of this, there is a high interest in using passive cooling techniques that are easily operated. Thus, the combination of conventional rehabilitation techniques together with the night ventilation cooling potential of the building's thermal mass was studied and quantified by using CFD simulations.

The results of this work validate the previous argumentation, since the optimal solution, if a traditional procedure had been followed, would imply a 70% reduction of the heating needs but also a 69% increase in the cooling needs. Consequently, there is a great need for considering passive cooling techniques. As mentioned before, the present study then analyzed how night ventilation requires the study of the techniques to activate the thermal mass. This activation was achieved by guaranteeing a high heat transfer between the fresh air and the walls of the building during cold hours at night. For that, an efficiency parameter was defined as indicator of the quality of that activation. This parameter was studied and quantified by using numerical techniques. The simulations that were presented allowed to analyze different possible designs as well as choose the most efficient one from the point of view of night cooling. Once the results of the numerical study were obtained, they were coupled with EnergyPlus simulations and the cooling demand reduction potential was quantified. The reduction that was achieved was even higher than 40%, with little additional costs.

From the present study, it may be concluded that it is possible to design innovative, integrated and economically feasible rehabilitation projects in historic buildings, and the solutions proposed in this study will be implemented and measured in the near future. The proposed methodology could be replicated in many of the historic buildings located in historical European city districts, since all of them present similar characteristics in terms of architectural constraints, construction technology, and existing energy systems. Last of all, the application of the proposed methodology in different climates (particularly those with severe summer conditions) would provide valuable outcomes.

**Author Contributions:** S.Á.D. and J.S.R. have developed the method and supervised the work. M.G.D. and J.A.T.R. have developed the BIM model in REVIT, google sketch up geometry and initial case in Energyplus. M.P.M. and L.R.R. have done all the simulations of the conventional catalog. M.P.M., J.S.R. and M.G.D. have worked in the Computational Fluid Dynamics (CFD)part. S.Á.D., J.S.R. and J.A.T.R. have analyzed and validated the results. All of them have written the paper.

**Acknowledgments:** The authors would like to take this opportunity to thank the DACAR project (Distritos de Balance Energético Nulo Mediante Algortimos de Confort Adaptativo y Gestión Óptimo de Redes Energéticas BIA2016-77431-C2-2-R), funded by Ministry of Economy and Competitiveness of Government of Spain (within the National Program for Research Aimed at the Challenges of Society, included in the framework of the Spanish National Plan for Scientific and Technical Research and Innovation) and European Regional Development's funds (ERDF) for its partial support. The authors also thank the University of Seville for its financial support through the US Research Plan VI (VPPI-US).

**Conflicts of Interest:** The authors declare no conflict of interest.

## Appendix A

In the following table, the materials of all the enclosures of the building under study are presented, together with their properties and thickness. This information allows the calculation of the thermal transmittance of all the elements of the building, which were used in Table 1 of the manuscript.

**Table A1.** Materials of the building's enclosures.

| Exterior Walls | | Interior Walls | | Roof | | Floor | | Window |
|---|---|---|---|---|---|---|---|---|
| Material | Thickness (m) | Material | Thickness (m) | Material | Thickness (m) | Material | Thickness (m) | Single Glass |
| Limestone | 0.36 | Plaster | 0.02 | Concrete | 0.2 | Stones | 0.03 | Wooden Carpentry |
| Brick | 0.11 | Brick | 0.08 | Plaster | 0.02 | Mortar | 0.04 | |
| Plaster | 0.02 | Plaster | 0.02 | | | | | |

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
