# Peer review of "Design of the Refurbishment of Historic Buildings with a Cost-Optimal Methodology: A Case Study"

_applsci, doi:10.3390/app9153104_

Round 1
Reviewer 1 Report
The authors addressed the cost-optimality search of retrofit interventions on a case study historic building. The topic is relevant and timely, but the quality of the paper is not enough for publication.
The authors should address the following major revisions as a consequence:
the title needs to amended to better reflect the contents of the paper (and be shortened as well)
l.40: please say this provision is for new constructions only
l.55: it is not lcear why the Cadiz location is that peculiar, please elaborate on this
double check reference in all the manuscript as there are missing ones
Fig. 1: it is not clear the difference between energy needs and consumption, please explain it better in the text
Introduction: I suggest referring the following paper concerning renewable energy exploitation in urban contexts: V. Costanzo, R. Yao, E. Essah, L. Shao, M. Shahrestani, A.C. Oliveira, M. Araz, A. Hepbasli, E. Biyik, A method of strategic evaluation of energy performance of Building Integrated Photovoltaic in the urban context, Journal of Cleaner Production 184 (2018), 82-91
Introduction should also better point out the aims and objectives of the paper at its end
Fig. 2 should also include the LCA to comprehensively depict the methodology adopted in the research
Table 1: it is important to list all the material layers for each construction component (e.g. walls), along with their thermophysical properties used in the simulation exercise
l. 295-299: if so, please report only Table 5 without mentioning all the possible combinations stemming from table 4 (to be removed as a consequence)
Table 5 proposes an ACH of 0.075: this is too low and unfeasible! Simulations using this value should be re-run using more realistic values. I do suggest to refer to this paper: V. Costanzo, R. Yao, T. Xu, J. Xiong, Q. Zhang, B. Li, Natural ventilation potential for residential buildings in a densely built-up and highly polluted environment. A case study, Renewable Energy 138 (2019), 340-353
Table 6: please change isolation with insulation
l.349-350. again, ACH of 12 and 16 are totally unrealistic, especially in urban context. Please refer to the paper suggested at point 11
results: main issue is that the optimal solution the authors claim to have found is not optimal. In fact, only if the total (heating and cooling together) energy needs are minimized, than we can talk about a cost-optimal solution. I do sugges the authors to re-write this section according to this comment
Discussion section is a Conclusions section in reality. please amend it accordingly
Author Response
Manuscript ID: applsci-556923
Title: Achievement of ZEB target on Historic Building by new approach to solve the problem of overheating using nightcooling
Dear Reviewer,
First of all, our sincerest thanks for your comments on our manuscript. We think they have unquestionably contributed to improving the quality of the paper since it helped us take into account some issues which we did not thoroughly consider.
· The manuscript has been modified to address each of the excellent comments and recommendations raised.
· The authors have included additional explanations.
· The 'tracked changes' tool has been used to make clear where revisions of the manuscript have taken place.
· English language and style has been revised in the whole manuscript.
We feel the revised manuscript has been greatly strengthened by the comments, and hope that this will comply with the remarks which were pointed out. We will now proceed to respond to each of them in detail.

Reviewer 2 Report
The paper deals with a parametric simulation activity aimed at defining the best solution for the energy refurbishment of existing buildings in order to reconvert them into nearly-zero energy ones. The paper documents well the research activities performed, although several changes are required before the paper will be ready to be published.
The major missing points are:
- Introduction: there are some references missing (e.g. lines 73, 83). Please double check the manuscript
- Methodology: The authors consider as a case of study an historic building (an XVIII century convent) and they model it in a commercial software. We do already know that the software is a reliable one as it has been used in several similar research activities, but the process folllowed to calibrate the variables of the building (e.g. U-values of walls, windows, roofs, ...) in order to have a model consistent with reality is not clearly described. It is essential to have a section on calibration before moving on with all further activities.
- On the same topic, it is not clear how values included in table 1 have been found. Do they derive from experimental activities (in this case the paper should briefly include the major results)?
- Section 2.4 conventional catalogue. It is not cleary how the catalogue has been defined. Does it includes a theoretical list of interventions? How the interventions are meant to be compatible with the historic value of the building? For example, I see that there is a proposal to reduce the thermal transmittance of walls to 0.3 W/m2K. I presume that for reaching a so low value, a thermal insulation layer has to be installed. Which technology is used and how is it compatible with the building? I was expecting to find such as these detailed information in the paper.
- Results and discussion are fine, although my major concern is the methodology applied to reach those results as highlighted in my previous comments.
Finally some minor points.
- The manuscript has to be carefully proof-read by an English native speaker, as it contains several language inaccuracies
- I have possibly found a mistake in the names of authors since the MCarmen Pavón Moreno is repeated two times
Author Response
Manuscript ID: applsci-556923
Title: Achievement of ZEB target on Historic Building by new approach to solve the problem of overheating using nightcooling
Dear Reviewer,
First of all, our sincerest thanks for your comments on our manuscript. We think they have unquestionably contributed to improving the quality of the paper since it helped us take into account some issues which we did not thoroughly consider.
· The manuscript has been modified to address each of the excellent comments and recommendations raised.
· The authors have added additional explanations.
· The 'tracked changes' tool has been used to make clear where revisions of the manuscript have taken place.
· English language and style has been revised in the whole manuscript.
We feel the revised manuscript has been greatly strengthened by the comments, and hope that this will comply with the remarks which were pointed out. We will now proceed to respond to each of them in detail.

Reviewer 3 Report
The paper presents research results from the rehabilitation of a building of the eighteenth century and turns it into a landmark building that will be used as a museum.
The paper contains several grammatical language errors and it needs second proofreading. I suggest the authors considering English proofreading by a third party. Some of the sentences are not clear. e.g. line 23-24. What is a road map? Not scientific! Please change to either Methodology or Process
The layout structure of this paper is suitable, but please provide a space between the text and tables or figures. e.g. table 1 or 2
The topic is not formulated appropriately. Too long, and many people may not know what ZEB means! I suggest the authors rephrase it. I also recommend using the word Rehabilitation or Retrofit in your topic.
Line 17, abstract – The abstract contains the required information, however it has not been executed sufficiently. It is a bit difficult to follow the text, and that I think is due to weak English expression.
Line 22, “However, this presents a significant issue...” – What issues? Please give some example in a bracket.
Line 23 - Please change the tense of the sentence to present: change “a road map has been” to “a road map is”
Line 25 – Same issue with the tense. See previous.
Line 25 – What is “BIM structure”? Although I know what you mean by that, it is difficult for a reader to understand it. This sentence needs to be restructured and a bit extended. It is not clear. You have raised the flag of BIM in this paper, but it is not addressed sufficiently! Perhaps, just change to a developed 3D model in Revit.
Line 34, Keywords – Please be careful about what you choose as the keywords! The word “Retrofit” is almost not used in the whole paper! Change to “Rehabilitation”. NZEB and LCC, please do not use abbreviation. Important note: check if you can find all your keywords in the abstract! This is also one way to improve the abstract!
Line 38, Introduction – It is too long and very difficult to follow. I suggest authors break it down in some subsections. e.g. Background, Statement of the problem, Research structure etc.
Line 48 - Please use the following paper to refer and support your discussion concerning sustainability.
Kamari, A., Corrao, R., & Kirkegaard, P.H. (2017). Sustainability focused Decision-making in Building Renovation. International Journal of Sustainable Built Environment. 6(2), 330-350. doi:10.1016/j.ijsbe.2017.05.001
Line 52 – What does Roadmap mean ? For who?
Lines 52- What do you mean by “Recovered” ? Not clear! Do you mean “Refurbished” or “Renovated” ?
Lines 72,73 – What do you mean? Not clear! Please extend and provide more information.
Line 73 – There is a referencing error.
Figure 1 – Not informative and can be removed. Are the figures yours?
Line 83 – There is a referencing error.
Lines 147-151. Try to use similar texts in the abstract and conclusion. This seems to be more understandable and better formulated.
Line 150 – Please change “particularities” to “specifications”.
Line 154 – Why you jump to the next page?
Line 161 – Please double check the caption.
Line 162 – Encompasses
Line 162 – I suggest creating a table and provide steps and their required descriptions.
Line 177 – The purpose of what? This paragraph does not seem to be linked properly.
Line 197 – What do you mean by “to define …”? Not clear.
Section 2.2 – In general it is not clear how you have developed the 3D model? Which technique did you use? Laser scanning? How accurate has been your model? What is the LOD of your model?
Line 226 – How it was done?
Line 256 – What is Building Demand?
Line 378, Results – It is reasonable
Line 463 – Please extend the caption. Not clear.
Line 500 – What does the first sentence mean?
Line 509, Discussion – Please change the word “Discussion” to “Conclusion”, as it seems there is no discussion in the text you have provided. The conclusion is still weak! Moreover, please provide the limitation of this study, and further research work regarding the topic. One way of developing such a work can be towards use of more innovative approaches to rapidly generate renovation scenarios. I suggest the authors use the following paper in this regard:
Kamari, A., Schultz, C., & Kirkegaard, P.H. (2019c). Constraint-based renovation design support through the renovation domain model. Automation in Construction. Vol. 104, 265-280. doi:10.1016/j.autcon.2019.04.023
Overall evaluation: Unfortunately, while many aspects of the research work are interesting and promise to bring value to the targeted area of research, the paper itself seems to be not clear enough as well as weak English expression. Above all, the research questions are not listed clearly. I encourage the authors to look after the comments and try to address them carefully. Thus, while the research is certainly worth presenting, major revision and resubmission is required, in my opinion.
Author Response

(The authors gave the same response as above.)

Round 2
Reviewer 1 Report
The authors carefully reviewed the manuscript as per the reviewers' comments.
I think in its actual shape the paper is ready for publication and no further amendments are needed.
Author Response
Manuscript ID: applsci-556923
Title: Achievement of ZEB target on Historic Building by new approach to solve the problem of overheating using nightcooling
Dear Reviewer,
First of all, our sincerest thanks for your comments on our manuscript. Although the assessment was positive, the entire document has been reviewed again, to reach a more desirable situation.
· The authors have additional explanations.
· The 'tracked changes' tool has been used to make clear where revisions of the manuscript have taken place.
· English language and style has been revised in the whole manuscript.

Reviewer 2 Report
The authors have replied to all comments and the paper is now ready to be accepted
Author Response

(The authors gave the same response as above.)

Reviewer 3 Report
The paper contains some grammatical language errors and it still needs second proofreading.
In addition, the conclusion is still weak! Moreover, please provide the limitation of this study, and further research work regarding the topic. One way of developing such a work can be towards use of more innovative approaches to rapidly generate renovation scenarios. Please use and cite the following paper as a potential for future work:
Kamari, A., Schultz, C., & Kirkegaard, P.H. (2019). Constraint-based renovation design support through the renovation domain model. Automation in Construction. Vol. 104, 265-280. doi:10.1016/j.autcon.2019.04.023
Author Response
Manuscript ID: applsci-556923
Title: Achievement of ZEB target on Historic Building by new approach to solve the problem of overheating using nightcooling
Thank you for your revision of our manuscript. We have included the mentioned reference, which was very interesting four our conclusions section, and emphasized the findings and conclusions of our study.
In addition, a native English speaker has revised again the whole manuscript, as can be seen in the manuscript version with control changes. We hope to have addressed all of your valuable comments satisfactorily, and we believe that the revised version has been greatly improved by them. Thank you again.